# What Nephrologists Should Know about the Use of Continuous Glucose Monitoring in Type 2 Diabetes Mellitus Patients on Chronic Hemodialysis

Faiza Lamine [1,2] , Menno Pruijm [3] , Virginie Bahon [4] and Anne Zanchi [2,3,4,*]

1   Unit of Diabetes and Endocrinology, Department of Internal Medicine, Riviera-Chablais Hospital (HRC), 1847 Rennaz, Switzerland
2   Service of Endocrinology, Diabetes and Metabolism, Lausanne University Hospital, 1011 Lausanne, Switzerland
3   Service of Nephrology and Hypertension, Department of Medicine, Lausanne University Hospital and University of Lausanne, 1011 Lausanne, Switzerland
4   Diabetes Unit, Clinique de la Source, 1011 Lausanne, Switzerland
*   Correspondence: Anne.Zanchi@chuv.ch

**Abstract:** Patients with type 2 diabetes (T2D) and end-stage kidney disease (ESKD) on renal replacement therapy represent a specific population with high morbidity and mortality, an increased risk of hypoglycemic episodes and large intra- and interdialysis glycemic variability. Antidiabetic treatment adjustment is therefore challenging, especially in insulin-treated patients. Continuous glucose monitoring (CGM) is increasingly proposed to T2D patients on hemodialysis (HD), although data regarding flash monitoring systems (FMSs) and real-time CGM (rtCGM) in HD patients are limited. Small CGM pilot studies of a short duration demonstrated improvements in glycemic control and decreased hypoglycemic events, despite a lower accuracy of CGM as compared to capillary blood glucose. Moreover, CGM–drug interactions with vitamin C, mannitol and paracetamol can occur in HD diabetic patients and need further study. Despite these shortcomings, professional CGM has the potential to become an integral part of glucose monitoring of HD patients treated with insulin. Personal CGM prescriptions can especially be useful in highly selected, motivated T2D HD patients on multiple daily insulin injections or experiencing frequent hypoglycemia with preserved diabetes self-management abilities or in whom diabetes is fully managed by medical providers. A close collaboration between the clinical staff working on HD units and diabetology teams, and ongoing patient education, are mandatory for optimal use of CGM.

**Keywords:** type 2 diabetes; hemodialysis; continuous glucose monitoring (CGM); hypoglycemia; hemodialysis-related glycemic variability; CGM accuracy; CGM–drug interactions

## 1. Introduction

Continuous glucose monitoring (CGM) is an innovative technology that revolutionized the management of diabetic patients over the past 15 years [1,2]. The first CGM marketed in 1999 (MINIMED CGMS®) contained a cable, lasted 3 days and required calibration with capillary blood glucose 2–4×/day. Since then, technological development have led to miniaturized, more precise, user-friendly, wireless CGM devices with no need for calibration and for some, the ability to connect to hybrid closed-loop insulin pumps [1].

As a consequence, the use of CGM devices has rapidly increased in patients with both type 1 and type 2 diabetes mellitus (T2DD). Patients with T2D and end-stage kidney disease (ESKD) on renal replacement therapy represent a specific population with high morbidity and mortality, an increased risk of hypoglycemic episodes and large fluctuations in glucose levels during and between dialysis sessions. Therefore, CGM is increasingly being proposed to T2D patients on hemodialysis (HD). As a consequence, nephrologists need to be updated on the optimal use and interpretation of CGM devices. This review

discusses the general principles, practical applications and possible caveats of CGM in dialysis patients. As data on the accuracy of therapeutic CGM in the setting of PD are not available yet, this review will only focus on their use in hemodialysis.

## 2. Types of CGM Devices

There are two types of CGM devices. The personal CGM is unblinded, and is intended for day-to-day use, including real-time CGM (rtCGM) and intermittently scanned CGM (is CGM, or flash monitoring system, FMS). The professional CGM devices are placed by caregivers, and provide retrospective data that are blinded or unblinded for a short period of time (10 to 14 days). Professional CGM is a useful diagnostic tool for identifying patterns of hypo/hyperglycemia and adjusting antidiabetic treatment.

Personal CGM improves glycemic control and prevention of hypoglycemia with possible improvement in quality of life [3]. The evidence on the efficacy of personal CGM is more robust in the context of type 1 diabetes (DT1), provided regular use of the device (>70% of the time), than in insulin-treated T2D. CGM is currently the standard for blood glucose monitoring in most patients with T1D, and is recommended early after the diagnosis of T1D in adults [4].

There is a growing demand for CGM devices in insulin-treated T2D. Despite the increasing availability of CGM in high-income countries, it is important to emphasize that optimal and safe use of personal CGM requires high initial investments from both patients and caregivers (interdisciplinary team specialized in diabetology) to ensure comprehensive and ongoing patient education and training. Prescribing personal CGM in patients with poor therapy adherence or altered self-management abilities is not recommended unless diabetes is entirely managed by caregivers trained in using CGM. It is also important to consider their relatively high cost and reimbursement issues. For these reasons, only diabetologists are authorized to prescribe personal CGM including FMS in Switzerland, provided that the patient is on multiple daily insulin injections or insulin-pump therapy.

Table 1 summarizes of the main technical characteristics and potential interferences with the four personal CGM devices most commonly used in Switzerland. Generally, CGM devices are not validated or approved for use in HD patients [5]. Caregivers and patients should be aware of their limitations, caveats and decreased accuracy in the setting of HD. Despite the lack of strong evidence, there is a general agreement that CGM use in selected insulin-treated diabetic patients undergoing chronic HD is a useful tool in combination with self-monitoring blood glucose (SMBG) and HbA1C to improve overall glycemic control and reduce hypoglycemic events [5–7].

**Table 1.** Key characteristics of the most frequently used personal CGM systems.

| | Dexcom | Freestyle Libre (Abbott) | | Eversense (Sensesonics) | Medtronic |
|---|---|---|---|---|---|
| | Dexcom G6® G7® | Freestyle Libre Freestyle Libre 2® | Freestyle Libre 3 | Eversense XL® | Medtronic Guardian Connect® (sensor: Enlite/Guardian sensor 3) |
| **Sensor technology** | Glucose oxydase | Glucose oxydase | Glucose oxydase | Fluorescence | Glucose oxydase |
| **CGM type** | Real-time (rt CGM) | Flash monitoring system (should be scanned at least every 8 h) | Real-time CGM | Real-time CGM | Real-time CGM |
| **Calibration (capillary glycemia) for maintaining sensor accuracy** | No | No | | Twice/day | Every 12 h |

**Table 1.** *Cont.*

| | Dexcom | Freestyle Libre (Abbott) | | Eversense (Sensesonics) | Medtronic |
|---|---|---|---|---|---|
| | Dexcom G6® G7® | Freestyle Libre Freestyle Libre 2® | Freestyle Libre 3 | Eversense XL® | Medtronic Guardian Connect® (sensor: Enlite/Guardian sensor 3) |
| **Sensor set up** | By the patient using an automatic applicator | By the patient using an automatic applicator | | Inserted under the skin/removed by a health care provider under local anesthesia | By the patient using an automatic applicator |
| **Insertion sites** | Abdomen and back of the upper arm Upper buttocks in children | Back of the upper arm | | Upper arm | Guardian 3: abdomen, upper buttocks, lower back, back of the upper arm Enlite: abdomen |
| **Sensor wear, days** | 10 days | 14 days | | 180 days | Guardian sensor 3: 7 days Enlite: 6 days |
| **Warm up period** | 2 h (G6), 30 min (G7) | 1 h | | 24 h | 2 h |
| **Receiver** | Dexcom-dedicated receiver/ compatible smartphone (Android or iOS) or SmartWatch | FSL/FSL2-dedicated receiver/ compatible smartphone (Android or iOS) | Compatible smartphone -Android: April 2022 -iOS: June 2022 (for Switzerland) | Compatible smartphone (Android or iOS), or Apple Watch | Guardian connect: compatible smartphone (iOS or Android) |
| **Memory storage of the sensor** | **N/A** | 8 h | **N/A** | None | NA |
| **Memory storage of the receiver** | 30 days (G6), 180 days (G7) (receiver), (90 days, in the cloud Dexcom Clarity) | 90 days | | Life of the transmitter (360 days) No limit in the cloud | 90 days (in the cloud, Carelink) 30 days in the pump |
| **Low/high glucose level alerts and alarm** | Yes | Yes for Freestyle Libre 2 | Yes | Yes | Yes |
| **Interferences with medical, X-rays or computed tomography (CT) scans or MRI \*** | All components must be removed before an X-ray or MRI, or (CT) scan, or high-frequency electrical heat (diathermy) treatments. | All components must be removed before an X-ray or MRI, or (CT) scan | | The transmitter must be removed before MRI or medical X-rays or CT scan procedure | All components must be removed in the presence of X-rays, CT, MRI or high-frequency electrical heat (diathermy) treatments |

**Table 1.** *Cont.*

| | Dexcom | Freestyle Libre (Abbott) | | Eversense (Sensesonics) | Medtronic |
|---|---|---|---|---|---|
| | Dexcom G6® G7® | Freestyle Libre Freestyle Libre 2® | Freestyle Libre 3 | Eversense XL® | Medtronic Guardian Connect® (sensor: Enlite/Guardian sensor 3) |
| **Device-drug interactions relevant for dialysis patients *** | Paracetamol (acetaminophen): falsely raises glucose readings if >1 g every 6 h Hydroxyurea or hydroxycarbamide may cause falsely elevated glucose readings (source: dexcom.com/interference) | Potential interference with vitamin C at high dosage (falsely raises glucose readings) Salicylic acid: falsely lowers glucose readings Source: https://www.freestyle.abbott/us-en/safety-information.html accessed on 26 May 2022 | | Tetracycline (falsely lowers glucose readings) • Mannitol or sorbitol, when administered intravenously, or as a component of an irrigation solution or peritoneal dialysis solution, may cause falsely elevated glucose readings | **Guardian 3:** Paracetamol (acetaminophen): falsely raises glucose readings) Fever reducer or cold medicine: falsely raises glucose readings Vitamin C (ascorbic acid) and Xylose can interfere with the meter: do not use the BG meter readings to calibrate the sensor **Enlite:** paracetamol (acetaminophen): falsely raises glucose readings) https://www.medtronic.com/us-en/healthcare-professionals/products/diabetes/indications-safety-warnings.html accessed on 26 May 2022 |

* Check updated recommendations from companies.

### 3. Why Are Diabetes Management and Glycemic Goal Attainment Challenging in Hemodialysis (HD) Patients?

Diabetes is the leading cause of renal failure (ESKD) and dialysis in high-income countries. Most patients with ESKD are on HD in the United States and Europe (>80% of ESKD patients). Among patients on HD, individuals with T2D have a higher mortality rate ranging between 50 and 60% at 5 years [8–10]. Uncontrolled diabetes in HD patients is associated with higher mortality. According to the Dialysis Outcomes and Practice Pattern Study (DOPPS study), which included 9200 HD patients with T1D and T2D, the association between HbA1C and mortality was U-shaped, with higher mortality with A1Cs <6.5% or >9% [11].

**Hyperglycemia:** Diabetic patients on chronic HD may have very high glucose levels despite relatively mild symptoms at presentation, with volume overload rather than volume depletion as osmotic diuresis is severely altered [12]. In the setting of diabetic ketoacidosis, clinical outcomes are worse in ESKD patients compared to patients with preserved renal function [13]. Lastly, severe hyperglycemia may compromise the course of a dialysis session (need to delay the start of the session and administer rapid-acting insulin for example) and

increase the risk of acute HD complications (cramps, intradialytic hypotension, intradialytic hypoglycemia, etc.).

**Hypoglycemia:** ESKD is associated with a very high risk of hypoglycemia with insulin or glinide therapies due to impaired renal glucose production, impaired hormonal counter-regulatory mechanisms and decreased renal clearance of hypoglycemic drugs. Moreover, many HD diabetic patients have impaired awareness of hypoglycemia due to long-standing diabetes. They will fail to recognize critically low glycemic episodes including nocturnal hypoglycemia if they only rely on SBGM. Hypoglycemia may increase cardiovascular morbidity (cardiac arrhythmia, stroke) and mortality (sudden cardiac death) and neurocognitive decline, especially in older patients [14].

**Glycemic variability:** Finally, HD patients have significant inter- and intradialytic glycemic variability that is hardly predictable and detectable by classic glycemic metrics (laboratory or point-of-care HbA1C and SMBG/capillary glycemia), even when using adequate glucose-containing dialysates [15,16].

Applying CGM in selected HD diabetic patients in the setting of a specialized and trained multidisciplinary care model is probably a useful tool for improving glycemic control while reducing hypoglycemic episodes, adjusting antidiabetic treatment, decreasing patient burden and also strengthening patients' therapeutic education. According to the latest KDIGO 2020 guideline on diabetes management in chronic kidney disease, CGM is an alternative approach to glucose monitoring in CKD including ESKD patients in whom HbA1C is less reliable [17]. This statement was shared by the KDOQI (Kidney Disease Outcomes Quality Initiative) work group for diabetes in CKD [6].

### 4. What Are the Limitations and Caveats of Conventional Markers of Glycemic Control (SMBG, HbA1C) and Alternative Markers (Fructosamine and Glycated Albumin) in HD Diabetic Patients?

HbA1C is recommended for the assessment of glycemic control in combination with SGBM in diabetic patients, including those with ESRD, as mentioned in the KDOQI clinical practice guidelines and in the latest guidelines from KDIGO [6,17,18]. It reflects average glycemia over approximately 3 months and it has a strong predictive value for diabetes complications. However, HbA1C does not provide a measure of glycemic variability or hypoglycemia frequency and is less accurate/reliable in the setting of ESRD [17,19]. Falsely low HbA1C values are frequently observed in hemodialysis patients owing to factors that increase red cell turnover in this setting (use of erythropoietin-stimulating agents, iron supplementation, reduced erythrocyte lifespan from uremia, erythrocyte lysis during HD sessions). Less frequently, falsely high HbA1C levels may be observed due to hemoglobin carbamylation secondary to elevated blood urea nitrogen and metabolic acidosis [15,16]. In clinical practice, analysis of the trend of HbA1C values is more informative than the analysis of the absolute value itself. A decreasing trend in consecutive HbA1C values for an individual HD diabetic insulin-treated patient, especially if HbA1C is <7%, should prompt caregivers to actively evaluate him for the occurrence of recurrent/asymptomatic hypoglycemia, which is a major issue in this population. Of note, burnt-out diabetes (HbA1C < 6%) can be observed in 30 to 40% of HD diabetic patients, prompting antidiabetic treatment discontinuation [20].

To overcome the uncertainty of HbA1C in ESRD diabetic patients, some experts suggest the use of alternative markers that reflect glycemia in a shorter time period (2–4 weeks), such as fructosamine and glycated albumin, for the monitoring of glycemic control in this setting. However, both marker assays are subject to bias (falsely low glycated albumin by hypoalbuminemia, falsely high fructosamine by hypoalbuminemia due to assay interference). Glycated albumin is not available in routine practice. Neither glycated albumin nor fructosamine has been sufficiently validated to support their use over HbA1c in advanced CKD [17]. In the research setting, the accuracy of HbA1c versus glycated albumin in detecting poor glycemic control among diabetic patients on hemodialysis was evaluated in a prior diagnostic test study that used 7-day-long CGM as a reference standard. Glycated

albumin was shown to be a stronger indicator of poor glycemic control assessed with 7-day-long CGM when compared to HbA1c [21].

CGM fully captures intra- and interdialytic glycemic variations and asymptomatic hypoglycemia, including nocturnal hypoglycemia, which cannot be detected by SMBG or A1C. Low/high-glucose-level alerts and alarms in real time, including predictive alarms, and trend arrows displayed on the screen of the CGM receiver or smartphone, allow patients to react immediately and appropriately to adjust their insulin dosing/food intake and prevent acute glycemic variations, especially hypoglycemia.

## 5. What Are the General Principles of CGM?

CGM is a convenient wearable device that uses an intradermal probe (sensor) to measure interstitial glucose in the subcutaneous adipose tissue every 1 to 5 min, with a time lag of 5–10 min over the actual plasma glucose. This lag is attributed to both a physiologic lag due to the time needed for attaining the equilibrium between the two compartments and a technical lag (signal processing) [22]. Of note, the time lag can be longer if glucose levels are increasing or falling rapidly (after eating, after hypoglycemia treatment, after insulin, following exercise), which can lead to discrepancies between CGM and SMBG readings. Thus, during periods of high glucose variability, treatment decision making should be based on SMBG results.

A transmitter placed on the skin is attached to the sensor or worn over the sensor, which transmits the glucose data to a receiver/smartphone via Bluetooth. CGM data are displayed on the screen of a receiver or a smartphone, either in real time or when the user actively scans the sensor. For the flash monitoring system, scans should be performed at least every 8 h to save all glycemic data.

Data displayed include trend arrows of rise or fall in glucose levels, adding context to the latest glucose readings [23]. Some CGM devices offer the option of personalized alarms for critical glycemia thresholds or critical glycemia variations, allowing the patient to make the appropriate treatment decision. Hypoglycemia alarms are very useful in insulin-treated patients known for hypoglycemia unawareness.

Currently used CGM systems are accurate, with an overall mean absolute relative difference (MARD) for SMBG ranging from 8.1 to 10.6%. [2]. The lower the MARD, the better the accuracy of the CGM device. Patients should be informed that CGM sensor accuracy is the lowest on the first day of use and slightly decreases at the end of the sensor's lifespan. SBGM is mandatory when symptoms do not match the CGM readings or in the case of unexpected/unexplained CGM readings. Additionally, CGM reading is less reliable if glucose levels are extremely low or high (<40 mg/dL-2.2 mmol/L or >400 mg/dL-22 mmol/L).

CGM–drug interferences can alter CGM accuracy. The main involved drugs are paracetamol (acetaminophen) and high-dose vitamin C, as both can interfere with glucose oxidase (used by all CGM devices except Eversense XL®), leading to falsely elevated glucose levels. According to the Dexcom G6® manufacture notice, paracetamol interference is limited provided drug exposure is less than 4 g/d. However, there are no data regarding paracetamol–Dexcom G6® interference in the setting of repeated exposure over several days [24,25]. Mannitol or sorbitol (intravenous administration) can falsely increase Eversense XL® readings while tetracycline can falsely lower Eversence XL® readings [26].

## 6. Interpretation of CGM: New Standardized Glucose Metrics

Guidelines for standardized CGM metrics were published in 2019 [27,28]. Key CGM metrics are summarized in Table 2.

**Table 2.** The 10 core CGM metrics.

| Number of Days CGM is Worn | Recommend 14 Days |
|---|---|
| **Percentage of time CGM is active** | Recommend 70% of days for 14 days |
| **Mean glucose** | |
| **Glucose management indicator (GMI), previously termed estimated HbA1C** | GMI can be used as a surrogate of HbA1C |
| **Glycemic variability (%CV) target** | ≤36% |
| **Time in range (TIR) target: % of readings and time per day 70–180 mg/dL (3.9–10.0 mmol/L)** | Type 1 and type 2 diabetes *: >70% (16 h 48 min)<br>Older/high-risk patients: >50% (>12 h) |
| **Time below range (TBR) target: level 1, % of readings and time per day 54–69 mg/dL (3.0–3.8 mmol/L)** | Type 1 and type 2 diabetes *: <4% (<1 h)<br>Older/high-risk patients: <1% (<15 mn) |
| **Time below range (TBR) target: level 2, % of readings and time < 54 mg/dL (3.0 mmol/L)** | Type 1 and type 2 diabetes *: <1% (<15 mn)<br>Older/high-risk patients: <0% |
| **Time above range (TAR): level 1 target, % of readings and time >181–250 mg/dL (10.1–13.9 mmol/L)** | Type 1 and type 2 diabetes *: <25% (<6 h)<br>Older/high-risk patients: <50% (<12 h) |
| **Time above range (TAR): level 2 target, % of readings and time >250 mg/dL (13.9 mmol/L)** | Type 1 and type 2 diabetes *: <5% (<1 h, 12 min)<br>Older/high-risk patients: <10% (<2 h, 24 min) |

* CGM-based targets specifically for pregnancy are more stringent.

Structured data analysis should include the following successive key steps:

- *Percentage of time CGM is active.* A minimum of 14 consecutive days of data with approximately 70% of possible CGM readings over those 14 days is mandatory for optimal analysis and decision making.
- *Glucose management indicator (GMI)*, previously termed estimated HbA1C: a metric derived from converting mean glucose (from CGM) into an estimate of concurrent (laboratory) glycated hemoglobin (HbA1c) using a population-based formula [29]. Discordance between GMI and measured HbA1C can be observed, and may be increased in advanced chronic kidney disease (eGFR < 60 mL/min) [30]. According to the 2020 KDIGO Clinical Practice Guideline on Diabetes Management in Chronic Kidney Disease, GMI can be used to evaluate glycemic control for CKD/ESKD individuals in whom HbA1c is unreliable [17].
- *Glycemic variability (% CV, coefficient of variation).* Increased CV is a marker of non-reproducible daily glucose curves, a pattern that is generally due to wide and recurrent glucose excursions, especially post-prandial hyperglycemia and/or recurrent hypoglycemia. Data that cannot be reproduced from day to day impact the possibility of adjusting the insulin regimen adequately and safely. Glycemic variability is correlated with risk of hypoglycemia and is an additional risk factor for diabetes complications [31,32]. The goal is a CV≤36%.
- *Time in range (TIR)*: % of readings and time per day 70–180 mg/dL (3.9–10.0 mmol/L). A TIR of 70% and 50% was shown to strongly correspond with an HbA1C of approximately 7% and 8%, respectively, in two cohorts of T1D and T2D. Recent studies support TIR as a predictive marker of diabetic microvascular complications [28].
- *Frequency and time spent in hypoglycemia* (time below range, TBR %): % of readings and time <54–69 mg/dL (3.0–3.8 mmol/L). Level 2, % of readings and time < 54 mg/dL (3.0 mmol/L)
- *Frequency and time spent in hyperglycemia* (time above range, TAR %): % of readings and time >181–250 mg/dL (10.1–13.9 mmol/L). Level 2, % of readings and time >250 mg/dL (>13.9 mmol/L)

- *Ambulatory glucose profile (AGP)* is a summary of the average daily glucose values by time of day over the report period (7-14-30-90 days), with median (50%) and other percentiles shown (Figure 1). This graph enables us to observe the pattern and the kinetics of hypo- and hyperglycemia.
- Daily glucose profiles.

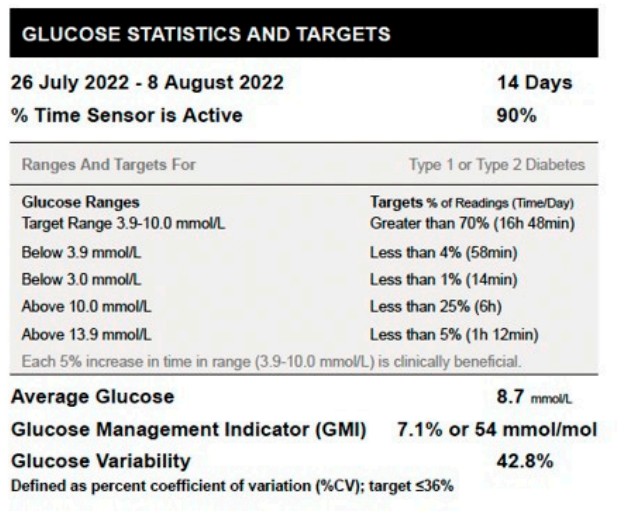
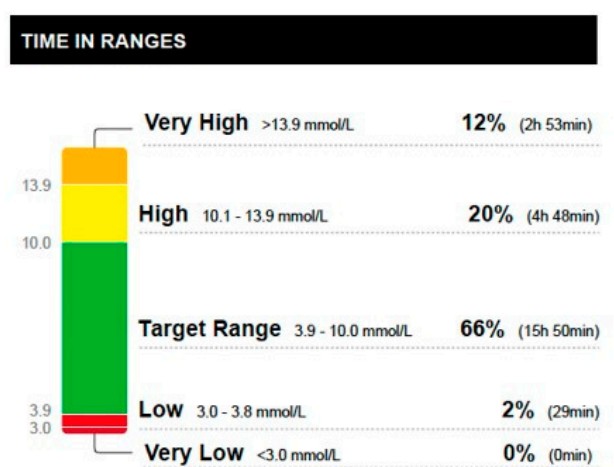
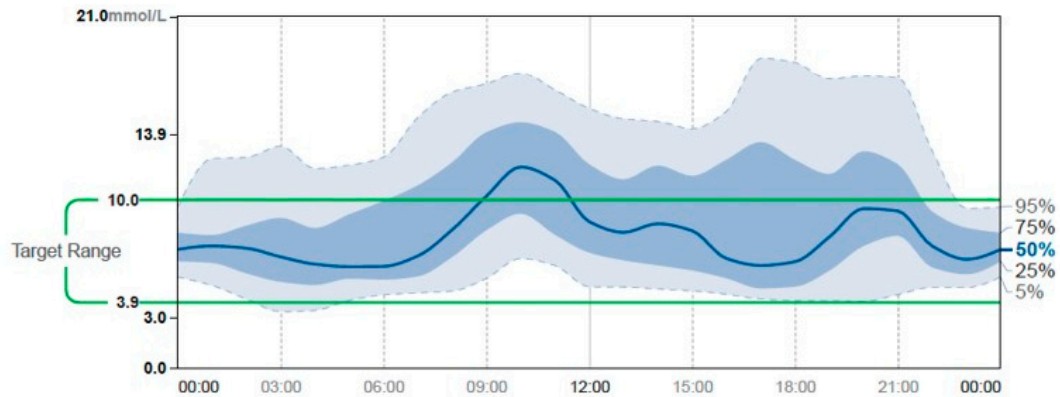
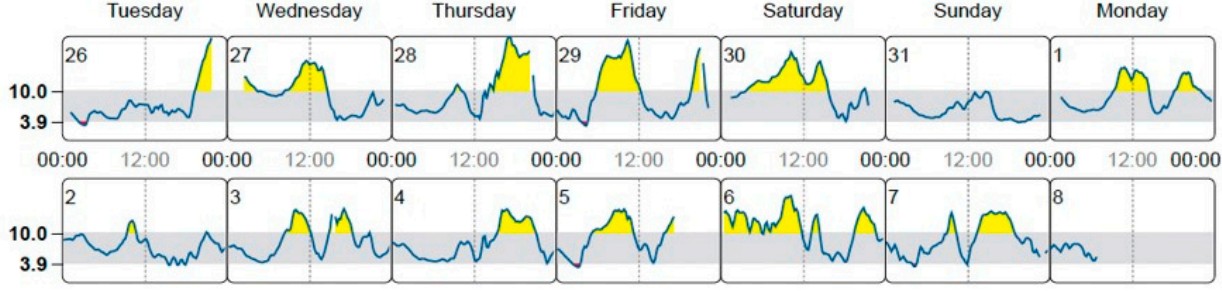

**Figure 1.** Ambulatory glucose profile [28].

These metrics are presented and visualized on the ambulatory glucose profile (AGP) report (Figure 1).

CGM-based glycemic targets are personalized. Stringent targets should be set for young patients while looser targets are recommended in older/frail patients and those with multiple comorbidities and a high hypoglycemic risk (Table 2). Although dedicated studies regarding optimal TIR for diabetic patients on dialysis are lacking, TIR cut-offs should be loosened as recommended for frail patients to decrease the risk of hypoglycemia.

### 7. What Have Studies Shown with CGM in the Setting of Chronic Hemodialysis?

Small pilot studies using professional CGM for short-term periods identified unique glucose variability patterns in diabetic patients (mostly T2D under insulin therapy) during and following maintenance HD sessions. They showed a substantial fall in glucose level, and a higher frequency of TBR (time below range) during and shortly after HD sessions, followed by reactive hyperglycemia and an increased TAR (time above range) typically occurring within the following day of HD [33–39]. Figure 2 illustrates the pattern of intra- and post-dialytic glucose variability.

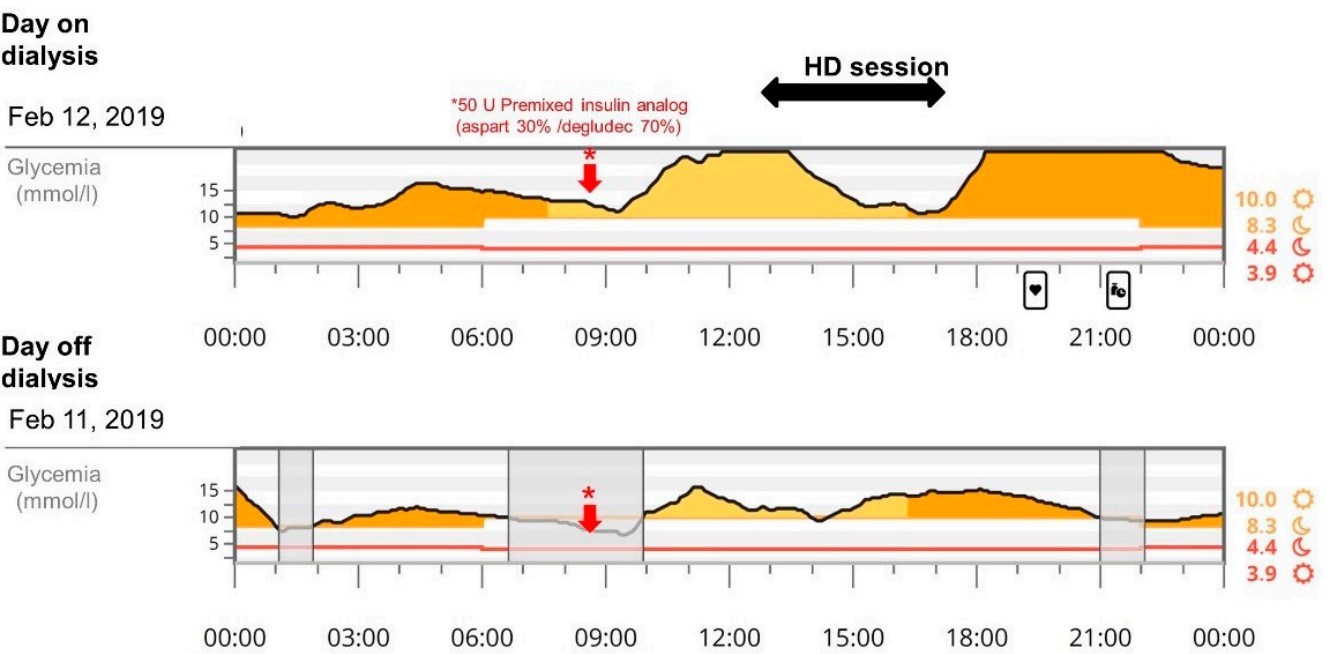

**Figure 2.** Intra and post-dialytic glycemic variability detected by the CGM (Dexcom G6) in a 82 year old insulin-treated type 2 diabetic patient: Major pre-dialytic hyperglycemia causing pronounced glycemia decrease during dialysis session followed by post-dialytic reactive hyperglycemia. On the other hand, the glycemic profile was stable during the previous day without hemodialysis.

In an observational study using CGM for 6 days in twelve T2D patients treated with multiple daily insulin injections, hyperglycemic peaks were observed on average 2.5 h after the end of the HD session, and were pre-prandial in 75% of cases [36]. In most studies, glycemic variability was significantly higher during the dialysis days than during the off-dialysis days [19,35,38–41].

HD-related hypoglycemia episodes are clinically relevant since they are frequent (16–23%) and often asymptomatic [34–37] and can occur with dialysates containing 100, 125 or 150 mg/dL glucose [42]. The frequency of hypoglycemic episodes was twice as high (4% vs. 2%) during dialysis days than off-dialysis days [40].

### 8. What Is the Role of CGM in Improving Diabetes Management in HD Patients?

The role of professional CGM use for short time periods in improving glycemic control in HD patients was illustrated by a limited number of small pilot studies. The use of

professional CGM (Navigator®, Abbott, Rungis, France) applied for 54 h at baseline and for a 3-month period in a group of 28 hemodialyzed patients with T2D treated by a basal–bolus detemir as part of an insulin regimen led to a significant decrease in HbA1C from $8.4 \pm 1.0\%$ to $7.6 \pm 1.0\%$ and a decrease in the TAR (>10 mmol/L) from $41.3 \pm 21.9\%$ to $30.1 \pm 22.4\%$ ($p < 0.05$), without a significant increase in the frequency of glucose values < 3.3 mmol/L. A CGM-adapted insulin regimen was performed by caregivers. Insulin requirements significantly increased from $70 \pm 51$ IU/d to $82 \pm 77$ IU/d, without significant changes in body weight [43].

In the DIALYDIAB pilot study that included 15 T2D patients on maintenance HD, remote analysis of CGM data (CGM iPro2®, Medtronic; Minneapolis; MN; USA) for 5 days repeated every 2 weeks for 6 weeks) led to more frequent insulin regimen adjustments by a single diabetes expert who gave therapeutic counseling to dialysis physicians. Additionally, fewer TBR episodes occurred as compared to the glucose metrics obtained during the initial SMBG (3 times a day) monitoring period [44].

Regarding the potential advantages of the long-term use of personal CGM, published data are extremely limited. A recently published case report illustrated the utility of transitioning from SBMG to CGM in terms of decreasing patient burden, improving glucose monitoring adherence, improving glucose metrics (especially less glycemic variability and increased TIR) and reducing hypoglycemia occurrence after 2 years of follow-up [45]. The patient included in this case report was relatively young (49 years) and very motivated to use CGM for the long term. Patient satisfaction with FSL use in the setting of HD was assessed in one small study that showed a very high patient satisfaction with FSL use during the 14-day period of the study considering its convenience (easy use) and usefulness despite the relatively advanced age of some among the 16 included patients (mean age 68+/−13 years) [46].

Taken together, despite the lack of strong evidence-based data, there is a general agreement that CGM use reduces TBR episodes in patients with T2D on HD and contributes to overall glycemic control; CGM may decrease patient burden when optimal patient education and a well-coordinated multidisciplinary/interdisciplinary care setting is provided. The Joint British Diabetes Societies and the Renal Association published a statement in favor of the use of CGM in dialysis patients in 2018, but also emphasized technical issues related to device calibration and reduced accuracy caused by rapid changes in blood glucose due to the dialysis process [7].

## 9. What Is the Accuracy of CGM/FSL in HD Diabetic Patients?

Data regarding FGM and rtCGM accuracy in diabetic patients undergoing chronic HD are limited. The majority of the studies were conducted with FSL, the most frequently used CGM worldwide [46–51]. Small studies have demonstrated that the accuracy of FSL is lower in diabetic HD patients, especially during HD sessions as compared to SBGM/capillary blood glucose. As stated above, the most commonly used metric for the assessment of CGM accuracy is the mean absolute relative difference (MARD). The MARD is calculated by averaging the absolute values of relative differences between FSL–CGM system results and corresponding comparison methods, mainly SBGM. A MARD of 10% or less for a CGM system is considered accurate enough for making insulin dosing decisions based on the CGM glucose reading without a confirmatory fingerstick. In one prospective study observing 16 patients with diabetes in a chronic HD program that analyzed 766 paired interstitial and capillary glucose levels, the MARD was 23%, and was higher during HD sessions (29%) over a 14-day period [46]. This finding was also reported in a smaller study that included 104 paired samples and used Freestyle Libre Pro® [49]. FSL seems to be less accurate for glucose values <10 mmol/L in HD diabetic patients, which increases the possibility of overestimating the hypoglycemia rate. The accuracy of FSL in patients undergoing HD deteriorates over time, with the global MARD increasing from 13.8% on day 1 to 21.6% on day 7 to 36.1% on day 14 [46]. The same finding was reported by Hissa et al. with MARD values ranging between 16.5 and 19% in the first week vs. 25.3–28.8% in

the second week [50]. The above-mentioned studies have also shown that FSL readings are significantly lower than SBGM readings in this population, but their correlation is well-established.

Only two small studies compared the accuracy between FSL (Freestyle Libre Pro®, (Medtronic, Northridge, CA, USA) and CGM (iPro2®, Abbott Japan, Chiba, Japan)) in patients with T2D undergoing HD [48,49]. Both studies found that FSL Pro was less accurate than CGM iPro2® at all times and during HD sessions in chronic HD type 2 diabetic patients. FreeStyle Libre Pro® was significantly less accurate than iPro2® for capillary glucose values between 3.9 and 10 mmol/L (70–180 mg/dL) (MARD 18.5% vs. 9.3%) but not for higher capillary glucose values between 10 and 13.9 mmol/L (180–250 mg/dL) (14.4% vs. 11,2%). The number of paired glucose measurements was insufficient in the hypoglycemic and level 2 hyperglycemic ranges for full analysis [49]. The accuracy of the Dexcom G6® in HD diabetic patients was evaluated in the *DEXCOM-HD study (NCT 04217161)*, which is ongoing.

The accuracy of Dexcom G6-Pro CGM (Dexcom, San Diego, CA, USA) in 20 diabetic outpatients on chronic hemodialysis was assessed in a recently published study, by comparing time-matched values to blood glucose measurements from the venous line (vBGM) during hemodialysis sessions and to SMBG values at home. The overall MARD was 13.8% for CGM- to SMBG-matched pairs and 14.3% for CGM- to vBGM-matched pairs during hemodialysis sessions. The reliability was analyzed with the Parkes error grid (PEG) with 86.7% of all SMBG pairs and all vBGM pairs in zone A (clinically accurate measurements) and with 98.7% and 100%, respectively, in zones A/B (clinically accurate or no risk from error). Of note, the reliability of Dexcom G6-Pro in the context of hypoglycemia could not be studied due to the lack of sufficient values in this range. Nevertheless, the authors pointed out that this device overestimated blood glucose readings in the majority of patients but without increasing the risk of under-diagnosis of hypoglycemia [52]

Mechanisms underlying the decreased accuracy of FSL and rtCGM in HD diabetic patients are still unclear. One hypothesis could be the hypervolemia-induced diluted glucose level in the interstitial tissue along with the rapid and marked changes in the fluid volume of the body, including subcutaneous interstitial tissue fluid caused by HD. The difference between capillary glucose and FSL readings was significant in patients who had ultrafiltration volumes of more than 2 L in the study of Hissa et al. [50]. However, no statistically significant correlation was found between FSL readings and the hydration status as measured by bioimpedance spectroscopy according to a dedicated small study [47]. The second hypothesis could be the wide and rapid intra- and interdialysis glucose variability. Other factors can influence the sensor performance and longevity, including foreign body response (FBR), which is an inflammatory reaction stimulated by the host's immune system in response to a foreign substance. Pro-inflammatory macrophages are recruited to the sensor site. This reaction significantly alters the accuracy of the sensor via augmented interstitial glucose consumption by the inflammatory cells at the sensor site. Moreover, fibroblasts are recruited and produce fibrosis that encapsulates the sensor, compromising glucose diffusion in the interstitial compartment [1]. Considering the broad range of cutaneous manifestations of ESKD [53], and the chronic and maladaptive inflammatory state associated with ESKD [54], altered FBR at the sensor site could be more pronounced in HD patients and could compromise CGM accuracy.

## 10. Conclusion and Practical Tips for Optimal Use of CGM in Type 2 Diabetes Mellitus Patients on Chronic Hemodialysis

Taken together, although CGM is not yet approved in patients on hemodialysis, it is increasingly used as an adjunctive tool to guide antidiabetic treatment adjustments in combination with HbA1C and SMBG.

In HD insulin-treated type 2 diabetes patients, the main target for frail subjects is to avoid hypoglycemic events, which are frequently asymptomatic and potentially severe and life-threatening. Small observational studies suggest that the number of hypoglycemic

episodes is reduced when CGM-based insulin management is used, but larger, randomized studies are needed.

Personal CGM prescription can be useful in highly selected, motivated HD patients with T2D on multiple daily insulin injections with preserved diabetes self-management abilities. For patients on insulin therapy without a personal CGM, professional CGM should ideally be performed as part of routine monitoring to ensure regular insulin dose adjustments and the detection of hypoglycemia. A structured AGP report analysis by a specialized diabetes management team may help clinicians customize the insulin regimen according to day-on /day-off dialysis. A close collaboration with the clinical staff working in HD units is mandatory.

Although data are lacking, it is generally advised that the sensor be inserted in the contralateral arm of the fistula. When using a non-factory-calibrated CGM, the CGM should be launched on a non-dialysis day to minimize calibration problems caused by rapid changes in blood glucose due to the dialysis process.

Initial and ongoing patient education and training regarding CGM device insertion, receiver and associated software is mandatory. However, it remains important to occasionally rely on capillary blood glucose as CGM accuracy may be impaired in dialysis patients, especially for glucose levels <180 mg/dL (<10 mmol/L) and in situations of wide glycemic intra- and interdialytic variability.

With future advances in technology, we are confident that the accuracy of continuous glucose monitoring will continue to improve for patients undergoing hemodialysis and become an essential tool for decreasing hypoglycemic risk and glucose excursions. We acknowledge that CGM devices are expensive, with rt CGM being approximately 3 times more expensive than FSL, and that there are disparities in access to this technology, especially in poor-income countries. Therefore, studies on the cost-effectiveness of CGM for hemodialyzed patients are needed.

Ongoing studies are also examining the accuracy of hybrid closed-loop insulin delivery, which relies on CGM readings. Preliminary results are promising and will, in our opinion, open a new chapter in glucose control in T2D patients on hemodialysis.

**Author Contributions:** Conceptualization, F.L. and A.Z.; methodology, F.L., A.Z. and M.P.; data curation, F.L. and A.Z.; writing—original draft preparation F.L.; writing—review and editing, F.L., V.B., A.Z. and M.P.; All authors have read and agreed to the published version of the manuscript.

**Funding:** This research received no external funding.

**Institutional Review Board Statement:** Not applicable.

**Informed Consent Statement:** Not applicable.

**Conflicts of Interest:** The authors declare no conflict of interest.

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
