# Peer review of "What Nephrologists Should Know about the Use of Continuous Glucose Monitoring in Type 2 Diabetes Mellitus Patients on Chronic Hemodialysis"

_kidneydial, doi:10.3390/kidneydial2030042_

Round 1
Reviewer 1 Report
The article covers an important topic and provides novel data. I have some additional suggestions to the authors:
1. HbA1c is used to determine the glycemic control status among diabetic patients on chronic dialysis, but this marker has several weaknesses. A brief description of the diagnostic and predictive value of this glycemic biomarker is needed.
2. The role of alternative glycemic biomarkers, such as the role of glycated albumin or fructosamine, needs to be discussed. In fact, these alternative biomarkers are promising, but still several factors limit their broad use in daily clinical practice. A new small section to cover these data will make your article stronger.
3. A new table to summarize the design and the main results of studies that evaluated the role of continuous-glucose-monitoring is also useful.
4. The accuracy of HbA1c versus glycated albumin in detecting poor glycemic control among diabetic patients on hemodialysis was evaluated in a prior diagnostic-test study that used 7-day long CGM as a reference-standard (Divani M, et al. Am J Nephrol 2018).
Author Response
Response to the Reviewer 1
The article covers an important topic and provides novel data. I have some additional suggestions to the authors:
- HbA1c is used to determine the glycemic control status among diabetic patients on chronic dialysis, but this marker has several weaknesses. A brief description of the diagnostic and predictive value of this glycemic biomarker is needed.
We thank the reviewer for this comment. In order to address this issue, we added the following paragraph to the revised version of the article (page 5, paragraph 4)
- What are the limitations and caveats of conventional markers of glycemic control (SMBG, HbA1C), and alternative markers (fructosamine and glycated albumin) in HD diabetic patients? (L125-L140)
HbA1C is recommended for the assessment of glycemic control in combination with SMBG in diabetic patients including those with ESRD in the KDOQI and latest KDIGO guidelines (6,17,18). HbA1C reflects the average glycemia over approximately 3 months and has a strong predictive value for diabetes complications. However, HbA1C is not a measure of glycemic variability or hypoglycemia frequency and is less accurate/reliable in the setting of ESRD (17, 19). Falsely low HbA1C values are frequently observed in hemodialysis patients owing to the increased red cell turnover (use of erythropoietin-stimulating agents, iron supplementation, reduced erythrocyte lifespan from uremia, erythrocyte lysis during HD sessions). Less frequently, falsely high HbA1C levels may be observed due to hemoglobin carbamylation secondary to elevated blood urea nitrogen and metabolic acidosis (15,16).
In clinical practice, it is more informative to analyze the trend of HbA1C values than its absolute value. A decreasing trend in HbA1C values in insulin-treated HD patients, should prompt caregivers to look for asymptomatic hypoglycemia especially if HbA1C is <7 %. Of note, burnt out diabetes defined by the spontaneous resolution of hyperglycemia and a HbA1C<6% can be observed in 30 to 40% of HD diabetic patients and when associated with protein-energy wasting is a sign of poor outcomes (20).
- The role of alternative glycemic biomarkers, such as the role of glycated albumin or fructosamine, needs to be discussed. In fact, these alternative biomarkers are promising, but still several factors limit their broad use in daily clinical practice. A new small section to cover these data will make your article stronger.
In the “new” paragraph 4, we added the following sentences (page 6, L141-L146):
To overcome the uncertainty surrounding HbA1C in ESRD diabetic patients, some experts encourage the use of alternative markers such as fructosamine and glycated albumin for the monitoring of glycemic control.
These markers have the advantage that they reflect glycemia over a shorter time period (2-4 weeks) However, the assays of both markers are subject to error, and may result in respectively a falsely low glycated albumin or falsely high fructosamine in case of hypoalbuminemia due to assay interference. Besides, glycated albumin is not available in routine practice in most countries, partly due to higher costs. Finally, neither glycated albumin nor fructosamine have been validated to support their use over HbA1c in advanced CKD (17).
- A new table to summarize the design and the main results of studies that evaluated the role of continuous-glucose-monitoring is also useful.
We thank the reviewer for this suggestion, but our article is already above the recommended 3000 word count. We prefer to refer to a recent review article on CGM by Gallieni M et al, Acta Diabetolo 2021(ref 33). This article includes a summary table with the main results of studies that evaluated the role of CGM. Therefore, we propose to add the following sentence to the beginning of the newly added chapter 8: What is the utility of CGM in improving diabetes management in HD patients?
The utility of professional CGM use for short time periods in improving glycemic control in HD patients was illustrated by a limited number of studies. A summary of the design and the main results of these studies is beyond the scope of this article, but was recently detailed by Gallieni et al (Ref 33)
- The accuracy of HbA1c versus glycated albumin in detecting poor glycemic control among diabetic patients on hemodialysis was evaluated in a prior diagnostic-test study that used 7-day long CGM as a reference-standard (Divani M, et al. Am J Nephrol 2018).
We thank the reviewer for drawing our attention to the publication of Divani et al. We discuss the main result of the study in the revised version, and have added the article to the reference list. The following sentence was added to the “new” chapter 4 to address this issue (L146-L149):
The accuracy of HbA1c versus glycated albumin in detecting poor glycemic control among diabetic patients on hemodialysis was evaluated in a prior diagnostic-test study that used 7-day long CGM as a reference-standard. In this study, glycated albumin was a stronger indicator of poor glycemic control than HbA1C. (new reference:21)
Reviewer 2 Report
To the Editor ok Kidney and Dialysis
It was with interest that I read Lamine’s manuscript “What nephrologists should know about the use of continuous 2 glucose monitoring in type 2 diabetes mellitus patients on chronic hemodialysis.
COMMENTS
the authors address a current issue such as glycemic control in the chronic patient on hemodialysis
The topic in always interesting due to the high incidence of diabetic patients on HD, characterized by higher glucose variability, known to be associated with increased mortality risk among diabetic patients on hemodialysis.
Various systems were studied to monitor glucose variability in HD patients: flash glucose monitoring (FGM) is a technology with considerable differences compared to continuous glucose monitoring (CGM) and self-monitoring of blood glucose (SMBG). The authors very well extensively describe the opportunity of some CGM systems in monitoring glycemic control in diabetics on hemodialysis treatment and describe a broad review of this technique, potentially capable of intercepting the wide glycemic variability in diabetic patients in HD.
The authors seem very optimistic about the possible use of GCM in the setting of these patients but then highlined the limitations of this method due to the low accuracy in diabetic HD patients, when compared to capillary blood glucose, a much more proven and certainly cheaper technique. The interaction with some drugs, namely paracetamol largely used instead of NSAIDs also in diabetics, could limit its possible use even more.
The utilization in diabetic patients on peritoneal dialysis in not sufficiently studied, a shame because in these patients, also due to the reabsorption of the glucose contained in the bags, a tight glycemic control is necessary also thanks to the greater continuity of PD treatment.
Finally, in table 2 the authors reported that glucose management indicator (GMI) can be used as a surrogate of HbA1C, but they do not explain why we must use a system (CGM) not validated, more expensive and with supposted lower accuracy than methods, such as the older HbA1c or other, namely FGM.
Author Response
COMMENTS
the authors address a current issue such as glycemic control in the chronic patient on hemodialysis
The topic in always interesting due to the high incidence of diabetic patients on HD, characterized by higher glucose variability, known to be associated with increased mortality risk among diabetic patients on hemodialysis.
Various systems were studied to monitor glucose variability in HD patients: flash glucose monitoring (FGM) is a technology with considerable differences compared to continuous glucose monitoring (CGM) and self-monitoring of blood glucose (SMBG). The authors very well extensively describe the opportunity of some CGM systems in monitoring glycemic control in diabetics on hemodialysis treatment and describe a broad review of this technique, potentially capable of intercepting the wide glycemic variability in diabetic patients in HD.
The authors seem very optimistic about the possible use of GCM in the setting of these patients but then highlined the limitations of this method due to the low accuracy in diabetic HD patients, when compared to capillary blood glucose, a much more proven and certainly cheaper technique. The interaction with some drugs, namely paracetamol largely used instead of NSAIDs also in diabetics, could limit its possible use even more.
We thank the reviewer for pointing this out. We certainly believe that CGM can be a very useful technique to monitor glycemic control in HD patients with diabetes, despite some technical issues. This is not only our view, but also the view of the Joint British Renal Societies and Renal Associations. With the revised version we modified the abstract and we added a paragraph on the strengths and weaknesses of the conventional measures of glycemic control HbA1C and SMBG (pages 5-6, paragraph 4). Finally, we added some advantages of CGM (page 6, lime 150-155).
We hope that the revised version provides a more homogenous, but realistic viewpoint on the actual state of knowledge and possible usefulness of this exciting new technique.
The utilization in diabetic patients on peritoneal dialysis in not sufficiently studied, a shame because in these patients, also due to the reabsorption of the glucose contained in the bags, a tight glycemic control is necessary also thanks to the greater continuity of PD treatment.
We fully agree that CGM use in the setting of peritoneal dialysis is an important issue which deserves more attention and specific recommendations due to the unique physiology of PD patients. However, due to limited space, we decided to focus solely on the use of GCM in chronic hemodialysis patients in this review.
Finally, in table 2 the authors reported that glucose management indicator (GMI) can be used as a surrogate of HbA1C, but they do not explain why we must use a system (CGM) not validated, more expensive and with supposted lower accuracy than methods, such as the older HbA1c or other, namely FGM.
We underline many advantages of CGM throughout the article, including their potential to detect hypoglycemia and glycemic variability related to dialysis sessions .We added the following sentence (L382-385)
We acknowledge that GCM are expensive. CGM being approximately 3 times more expensive than FSL, and there are disparities in access to this technology especially in poor-income countries. Therefore, cost-effectiveness studies of CGM use in hemodialyzed patients are needed. This issue was added to the revised version of the article.
Round 2
Reviewer 1 Report
No further comments.
Reviewer 2 Report
I appreciate the author's corrections and explanations to the reviewers' suggestions. To my opinion now the manuscript run better